# CIEASR: Contextual Image-Enhanced Automatic Speech Recognition for Improved Homophone Discrimination

## ABSTRACT

Automatic Speech Recognition (ASR) models pre-trained on large-scale speech datasets have achieved significant breakthroughs compared with traditional methods. However, mainstream pre-trained ASR models encounter challenges in distinguishing homophones, which have close or identical pronunciations. Previous studies have introduced visual auxiliary cues to address this challenge, yet the sophisticated use of lip movements falls short in correcting homophone errors. On the other hand, the fusion and utilization of scene images remain in an exploratory stage, with performance still inferior to the pre-trained speech model. In this paper, we introduce CIEASR (Contextual Image-Enhanced Automatic Speech Recognition), a novel multimodal speech recognition model that incorporates a new cue fusion method, using scene images as soft prompts to correct homophone errors. To mitigate data scarcity, we refine and expand the VSDial dataset for extensive experiments, illustrating that scene images contribute to the accurate recognition of entity nouns and personal pronouns. Our proposed CIEASR achieves state-of-the-art results on VSDial and Flickr8K, significantly reducing the Character Error Rate (CER) on VSDial from 3.61% to 0.92%.

## CCS CONCEPTS

• **Computing methodologies → Speech recognition**.

## KEYWORDS

Multimodal speech recognition, Multimodal fusion, Homophone discrimination

## 1 INTRODUCTION

Automatic Speech Recognition (ASR) systems are designed to transcribe spoken utterances into text with precision and reliability. Pre-trained speech models, including Wav2Vec 2.0 [4] and Whisper [37], which leverage large-scale speech datasets, have demonstrated remarkable performance in ASR tasks, exhibiting high recognition accuracy and maintaining robustness across various scenarios. However, these methods lack the functionality to utilize visual contextual information, making it challenging to achieve high precision with closely pronounced words or homophones. As demonstrated in **Figure 1**, semantically valid homophones can neither be distinguished by pronunciation nor be corrected by language model

*ACM MM, 2024, Melbourne, Australia*
© 2024 Copyright held by the owner/author(s). Publication rights licensed to ACM.
ACM ISBN 978-x-xxxx-xxxx-x/YY/MM
https://doi.org/10.1145/nnnnnnn.nnnnnnn

decoders. It is challenging to correct homophone recognition errors using purely auditory information.

**Figure 1: Introducing contextual images into the ASR system plays a significant role in correcting homophone errors. The blue words represent recognition confusion caused by homophones or near homophones, while the red indicates the correction results using scene images.**

Previous studies have demonstrated the potential benefits of integrating visual cues such as lip movements [41, 49] and gestures [29] into ASR systems. They are suitable for enhancing speech recognition systems' noise robustness and recognition performance in scenarios such as multi-person meetings and intelligent cockpits [8, 32, 54]. However, these methods require strict synchronization of timestamps, posing a relatively challenging requirement for real-world applications. Because relying solely on lip movements allows for the transcription of spoken content into text [10, 12, 27], such visual cues essentially act as a visual representation of speech, rather than offering visual semantic information. Therefore, these approaches remain inadequate for tackling the challenge of homophone discrimination.

A viable solution leverages semantic-level visual context information to supplement and enhance referential information missing in speech, thereby clarifying the actual meaning of homophones. Previous work has shown that contextual images can enhance the robustness and accuracy of ASR systems. Srinivasan et al. [43] adopts the Faster-RCNN model [38] to obtain fine-grained visual representations and conducts a detailed analysis of the alignment between visual and auditory elements, confirming the reliance of multimodal speech recognition models on relevant visual contexts during prediction. Ni et al. [31] utilizes context-dependent visual and corresponding linguistic cues to correct homophone errors distinguishable from scene images. However, their methods of integrating cues, predominantly through concatenation [33], attention mechanisms [43] or Multi-Layer Perceptron (MLP) fusion modules [31], are in a developing stage of utilizing contextual cues. These methods do not yet match the recognition performance of the cue-less speech pre-training model Whisper in terms of recognition capabilities.

To effectively address the challenge of correcting homophone errors, it is essential to better utilize the semantic information within scene images. Inspired by the multimodal alignment strategies of Multimodal Large Language Models (MLLMs) [6, 20, 23], we integrate scene images as soft prompts, diverging from the traditional methods of integrating cues. Specifically, we utilize the Q-Former [20], an efficient image-text alignment model, to extract semantic information from scene images which provides richer content than textual prompts such as key phrases, captions, and descriptions. To address the scarcity of tri-modal alignment data among images, speech, and text, we revise and expand the VSDial dataset. Furthermore, we exploit the prompt interface of the Whisper model to integrate image-derived soft prompts in a seamlessly natural manner.

The performance of our proposed model demonstrates on both the synthetic dataset **VSDial** and the real dataset **Flickr8K**, with speech recognition error rates decreasing from 3.61% and 2.42% to an impressive low of **0.92%** and **2.05%**, respectively. We indicate that when it comes to speech recognition involving homophones, the relevant entity information provided by contextual images can accurately identify key vocabulary within the speech. Our contributions are summarized as follows:

- We introduce CIEASR, a novel multimodal ASR model that enhances the contextual visual comprehension of ASR systems, offering an effective solution to the challenge of homophone discrimination.
- We propose a new method of cue integration, incorporating scene image cues as soft prompts under the pre-training paradigm which yields significant improvements in the recognition of entity nouns and pronouns.
- We revise and expand the VSDial datasets and conduct comprehensive experiments on both VSDial and Flickr8K, demonstrating the significance of scene images in ASR systems. CIEASR achieves new state-of-the-art results on these datasets.

## 2 RELATED WORK

### 2.1 Pre-trained Speech Models

The field of speech recognition significantly progressed with the introduction of unsupervised pre-training techniques such as Wav2Vec 2.0 [4], HuBERT [17], w2v_BERT [11], etc. The unsupervised pre-trained speech models analyze vast amounts of data to capture the underlying patterns of speech. However, they require an additional fine-tuning step due to the lack of a decoding mapping to useful outputs. Whisper [37] is a speech recognition model pre-trained in a weakly supervised fashion across large-scale speech datasets. Goron et al. [14], Moor et al. [30], Zhang et al. [53], among others, adopt Whisper as a backbone for downstream tasks due to its flexible Encoder-Decoder architecture and robust speech pre-training capabilities. In this work, Whisper is adopted as our foundation model to obtain strong speech recognition performance.

### 2.2 Cue-enhanced Speech Recognition

Traditional speech recognition methods have primarily focused on robustly and effectively transferring speech to text using only auditory inputs [15, 16, 47, 50]. The integration of multimodal cues into ASR systems has been identified as a potent means to enhance robustness and accuracy [2]. Commonly used cues include lip movements [28, 34, 40, 41, 49] and gestures [29]. Notably, lip movements require precise synchronization of video frames and speech segments to be effective [35], and cannot adequately address the problem of correcting homophone errors. Recent studies have increasingly investigated the application of readily accessible visual contexts, including images and videos, to enhance recognition performance [13, 26, 31, 36, 42, 43]. Srinivasan et al. [42] found that visual context can help ASR systems recover masked speech of objects by using images as auxiliary signals. Based on this discovery, Srinivasan et al. [43] automatically detects object proposals and directly grounds speech into regions of an image, Ma et al. [26] adopts a self-supervised pre-trained text-video embedding model to extract visual information and improve the recognition of specific words. Other applications include extracting visual context by a visual detector for embodied agents[36] and exploring various cross-modal fusion schemes to combine visual and linguistic information to make speech recognition more accurate and versatile [31].

### 2.3 Multimodal Fusion and Alignment

Multimodal models, such as GPT-4V [1] and Gemini [46], have demonstrated considerable versatility across diverse fields. Existing multimodal models primarily leverage Visual-Language Pre-training (VLP) techniques to align vision and language representations [7]. Typically, VLP models that emphasize multimodal understanding comprise three components: Modality Encoder, LLM Backbone, and Input Projector which transforms encoded visual features into linguistic embedding space [52]. This methodology is adaptable to other modalities, such as audio and video, through modifications to the encoding mechanism [9, 22, 51]. LLaVA [23], Qwen-VL [5], Qwen-Audio [9], among others, use an MLP to map encoded features to textual feature space. Flamingo [3] fuses compressed input feature sequence with text through cross attention. Q-Former [20] extracts relevant features from the encoded features with fixed-length learnable queries and treats the selected features as cross-modal prompts, thereby enhancing the model's interpretative and generative capabilities.

## 3 METHOD

### 3.1 Model Architecture

The CIEASR model is a novel approach for addressing homophone errors in speech recognition by leveraging the contextual information of scene images. It incorporates scene image cues as soft prompts into the speech recognition framework, distinct from traditional fusion approaches.

As illustrated in **Figure 2**, raw images are encoded by a **Visual Encoder**. Encoded image features are distilled and condensed into high-level semantic features through a **Q-Former**. These semantic features are aligned with the textual prompt interface space of the Whisper model through a projection layer, enabling the speech recognition system to gain nuanced insights derived from images without fine-tuning the Whisper model itself. This approach maintains Whisper's multilingual and robust capabilities while integrating image understanding. Specifically, the handling of scene

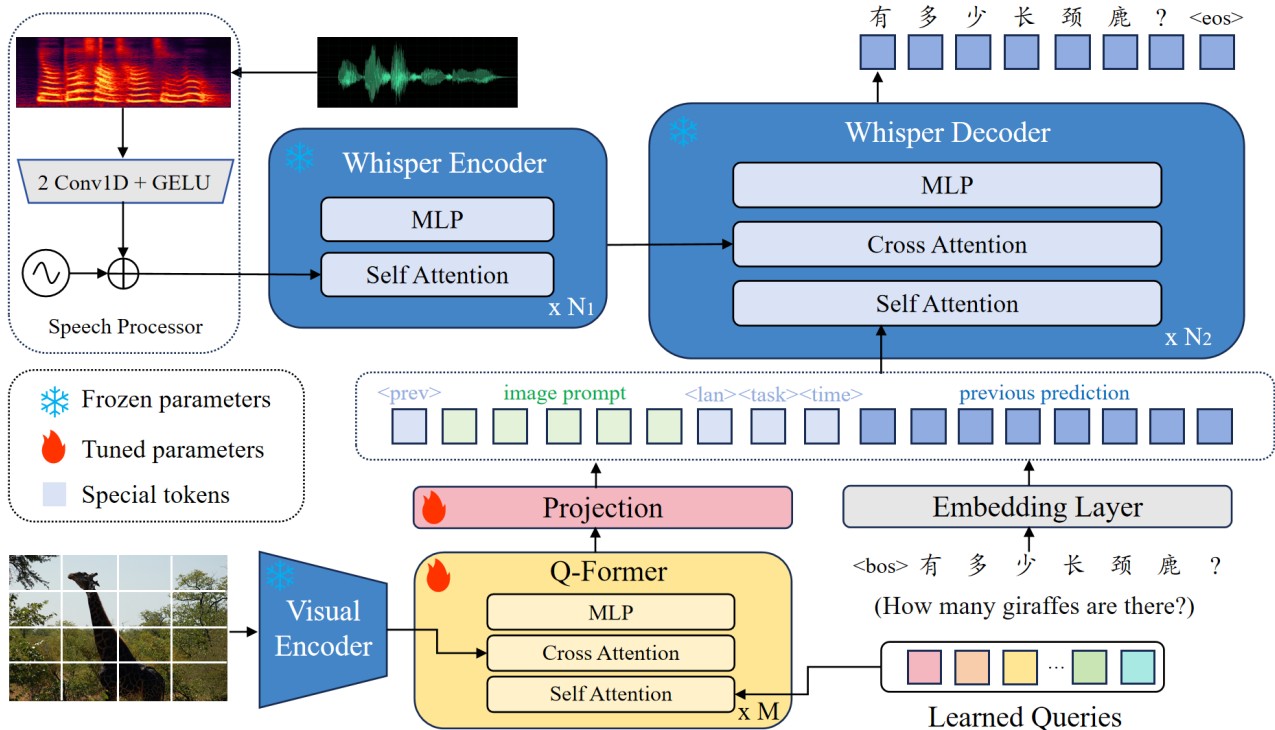

**Figure 2: The overall architecture of CIEASR:** The scene image is transformed into an image prompt through a Visual Encoder, a Q-Former, and a projection layer, which is then concatenated with the text embedding before entering the Whisper Decoder while speech is processed through a Speech Processor and Whisper Encoder, interacting with the image and text information in the form of cross attention.

images via Q-Former is presented in **section 3.2** while the fusion strategy and the speech decoder are presented in **section 3.3**

## 3.2 Contextual Images Alignment Strategy

To enhance the speech decoder's capacity for effective interpretation of image information, we employ the Q-Former [20] for the strategic extraction of semantic visual features. This approach aligns encoded visual representations seamlessly with the textual space. The training of Q-Former is divided into two stages:

*3.2.1 Representation Learning Stage.* The first phase involves image-text contrastive learning, where images $i$ are transformed into fixed-length image embeddings $E_i$ by a visual encoder EVA_CLIP [45]. $E_i$ interact with learnable queries $Q_i$ through cross attention. Training is conducted using image-text pairs with a contrastive loss, which includes Image-Text Contrastive Learning (ITC), Image-grounded Text Generation (ITG), and Image-Text Matching (ITM). The training in the first phase effectively extracts semantic information from images and compresses the length of the image embeddings.

*3.2.2 Generative Learning Stage.* After extensive training with paired image-text data endows the Q-former with the ability to extract image representations, Q-Former is linked to a frozen Whisper decoder through a fully connected layer to establish a bridge

between the image space and the textual space of the Whisper decoder. At this stage, our model performs autoregressive predictions of textual labels Y corresponding to both speech and image inputs. The detailed Speech Decoding process is presented in **Section 3.3**.

The addition of visual context $X_I$ from Q-Former significantly enhances the decoder's understanding of semantic content within speech signals, bridging the gap between visual inputs and textual transcriptions. This integration approach ensures the generation of textual labels coherent with both the acoustic information and visual context, leading to improved accuracy and robustness in speech recognition tasks.

## 3.3 Infusing Visual Cues into Speech Decoding

In large-scale pre-trained models employing autoregressive decoding, the critical function of text prompts in determining model outputs has been well-documented. Research in prompt engineering [18, 19, 48] demonstrates that meticulously designed prompts can significantly enhance the performance of generative models. Drawing on these insights, our study explores using contextual images as soft prompts in the Whisper framework, aiming to harness their potential to refine the model's output.

Though Whisper may not strictly be a Large Language Model (LLM), its autoregressive decoding aligns with LLM functionalities,

endowing it with substantial inferential capabilities. With scene image cues provided by Q-Former, the Whisper decoder essentially assumes the role of a language model, possessing capabilities for inference and error correction. It produces textual predictions $Y$ that mirror speech transcriptions $S$, utilizing image semantics $X_I$ as prompts for enriched speech transcription.

The origin speech $s$ is first transformed into a log-magnitude Mel spectrogram and then preprocessed by the Speech Processor, detailed in **section 4.2**. Then the processed speech signal $S$ is extracted to a series of acoustic features $X_S$ by the Whisper Encoder, which are then fed into the decoder using cross attention. This attention mechanism allows for a seamless blend of auditory and visual data in generating speech transcriptions. We have adapted the prompt interface of the Whisper decoder, shifting from text token IDs to image embeddings $X_I$. Utilizing the vocabulary of Whisper, we convert text label IDs into text embeddings $Y$, allowing them to be concatenated with $X_I$.

For a text label $Y$ with a length of $L$, the decoding process of the Speech Decoder can be represented by the equation (1):

$$P(Y \mid X_S, X_I) = \prod_{j=1}^{L} P_{\text{decoder}}(y_j \mid X_S, X_I, X_{sp}, Y_{i<j}) \quad (1)$$

In this equation, $Y$ represents the final output transcription text. $X_S$ denotes the speech features obtained from the Speech Encoder, while $X_I$ refers to the aligned image features derived from the Visual Encoder and Multimodal Alignment Module. $X_{sp}$ represents special tokens controlling the model's output, including language specification, task definition, and whether to use timestamps. During the decoding process, $X_S$ is inputted into each decoder block through cross-attention mechanisms. The special tokens $X_{sp}$ and text embeddings generated from previous outputs $Y_{i<j}$ are concatenated with the image representations $X_I$ and jointly fed into the Speech Decoder module for decoding.

## 4 EXPERIMENTS

### 4.1 Datasets

We conduct our experiments on the multimodal dataset **VSDial** and **Flickr8K**.

**VSDial** is a synthetic dataset consisting of 120,000 images. Each sample includes one image, its English description, and ten rounds of dialogue surrounding the image. The images are sourced from COCO and VisDial, while the synthetic speech is generated from VSDial using Fairseq to convert each image's corresponding ten segments of text questions into speech. The image caption serves as a linguistic cue for speech recognition. We have expanded the synthetic speech to include both questions and captions. We develop two versions of the caption in Chinese and English. Using **Paddle-Speech** for multi-speaker voice synthesis, we address the issue of the original VSDial dataset neglecting English words interspersed within Chinese sentences, ensuring comprehensive language processing. The expanded dataset **VSDial-caption** is set to be accessible for download under an open-source license, facilitating broader research utilization. For distinction, we refer to the original VSDial set as **VSDial-question**.

**Flickr8K** is a real speech dataset, serving as a subset of the larger image corpus Flickr30K, wherein captions are articulated by human voices. It encompasses 8,000 images, each associated with five unique English-recorded captions.Flickr8K totally contains a total of 30,000 audio samples for training and 5,000 samples each for validation and testing.

### 4.2 Experimental Setup

We conduct experiments on the original dataset **VSDial-question**, our expanded dataset **VSDial-caption**, and the real speech dataset **Flickr8K**.

For every ten pairs of questions and answers of an image in VSDial-question, only one question is randomly used, resulting in a total of 120,000/2,000/8,000 training/validation/testing samples, respectively. For the Flickr8k dataset, one corresponding speech sample is randomly selected for each image, resulting in a total of 8,000 samples for one training epoch.

We conduct experiments utilizing **VSDial-caption** across three linguistic settings: Chinese, English, and a random selection of Chinese and English speech utterances which is analyzed in section 4.4.

In practical training, we employ a ViT model with a resolution of $224 \times 224$. We initialize a Q-Former model which has completed stage one training, and the Whisper large-v2 model, with both the ViT and Whisper models being frozen. Consequently, our model is highly lightweight, with a parameter count of 1.0B when using CIEASR and 40k when using learned queries alone.

For speech preprocessing, speech utterances are uniformly resampled to 16,000 Hz and transformed to a log-magnitude Mel Spectrogram, computed using 25ms windows with a stride of 10ms. Inputs undergo global scaling to ensure they range between -1 and 1, achieving an approximately zero mean across the pre-training dataset.

### 4.3 Results on VSDial-question

Generally, for speech in logographic languages such as Chinese, researchers pay more attention to the Character Error Rate (CER), whereas for alphabetic languages like English, the Word Error Rate (WER) is of greater concern. We provide both metrics for reference.

**Table 1: Main WER(%) and CER(%) results on VSDial-question. Text-cn/en means using Chinese/English image captions as text prompts.**

| Model | Language | Cues | WER ↓ | CER ↓ |
|-------|----------|------|-------|-------|
| VILAS[31] | cn | image | / | **4.40** |
| VILAS[31] | cn | image+text-cn | / | 4.70 |
| WHISPER | cn | none | 3.81 | **3.61** |
| CIEASR | cn | image | 1.02 | **0.92** |
| CIEASR | cn | text-cn | 2.80 | 2.41 |
| CIEASR | cn | text-en | 3.47 | 3.31 |
| CIEASR | cn | image+text-cn | 1.20 | 1.02 |

As shown in **Table 1**, we can conclude from the Chinese question experiment that:

- Performance improvement with the introduction of contextual images: The introduction of contextual visual cues results in a significant performance enhancement. Compared to the pre-trained ASR model Whisper, our performance has improved by approximately **2.7** percentage points of CER. This proves that the integration of cues from other modalities can greatly enhance the performance of speech recognition systems.
- Effects of the introduction of contextual images on recognition: Introducing images enables the recognition of gender and entity information, including actual items present and other entities suggested by the scene. For instance, when given an image of an office scene, visual contextual information can provide details about items directly appearing in the image, as well as entities related to the office scene that are not present in the image.

In speech recognition systems, both scene image cues and textual cues offer vital contextual information. Textual cues within ASR systems deliver a focused subset of semantic information. In our experiment, image captions serve as highly pertinent textual cues for speech utterances linked to the respective image, thus qualifying as textual cues. Moreover, scene image cues not only furnish semantic information exceeding that provided by textual cues, encompassing aspects like background, texture, and spatial relationships but also reduce the high costs associated with manually designing textual cues in speech systems.

Upon examining the differences between scene image cues and textual cues, we find that scene images offer abundant visual information. This wealth of information further enhances automatic recognition performance, specifically resulting in a **1.5** percentage point improvement of CER compared to using textual cues in the same language.

In an analysis comparing the use of textual cues across various languages, we observe that employing Chinese textual prompts leads to an improvement in the performance of Chinese speech recognition. In contrast, the effect of cross-lingual textual prompts appears to be less significant. Quantitatively, utilizing textual cues in the same Chinese language enhances performance by **1.2** percentage points of CER, whereas different English language cues lead to a mere 0.3 percentage point increase, underscoring a limitation of textual cues in a cross-lingual context.

Furthermore, a common phenomenon is observed. In both our model and the VILAS model, the performance of simultaneously using both image and textual cues tends to be slightly lower compared to using scene image cues alone. This may be attributed to the scene image encapsulating the semantics of the textual cues described in the image, rendering the textual cues redundant in this context. However, the potential for further enhancing ASR system performance by simultaneously using scene images and textual cues that provide non-overlapping semantic information is an area left for future research in our work on the field of multimodal complementary fusion.

**Table 2: Main WER(%) and CER(%) results on VSDial-caption. Language "cn+en" indicates that a random speech segment is selected from the corresponding Chinese and English speech for training and testing.**

| Model | Language | Cues | WER ↓ | CER ↓ |
|---|---|---|---|---|
| WHISPER | cn | none | 3.04 | 2.51 |
| CIEASR | cn | image | 1.42 | **1.22** |
| WHISPER | en | none | 4.24 | 1.80 |
| CIEASR | en | image | **2.12** | 0.78 |
| WHISPER | cn+en | none | 3.61 | 2.13 |
| CIEASR | cn+en | image | **1.93** | **1.10** |

## 4.4 Results on VSDial-caption

As shown in **Table 2**, the following conclusions can be drawn:

- From the Chinese experiments, it can be deduced that due to the dataset's characteristics that the translated captions contain a significant number of numerals and phrases, the performance with the introduction of images is slightly lower than in the question experiments. However, there is still a considerable improvement of **1.3** percentage points of CER.
- The performance in the English experiments meets expectations, aligning with the higher WER and lower CER typically observed in English sentence recognition, with an enhancement in performance upon introducing images; there is a **2.2** percentage point improvement in WER.
- We also add mixed-language experiments and find that the performance of the ASR system lies between the Chinese and English experiments. This suggests that utilizing the Whisper multilingual pre-trained model as the backbone for the ASR system provides certain robustness to mixed languages.

In conclusion, contextual images, as a form of visual cue, can effectively enhance the entity information recognition capability of pre-trained language models, all without the need for any fine-tuning and efficiently improving system practicality with low computational resources. Compared with textual cues, image cues, as a denser form of information, can provide more context knowledge independent of text, and image information does not have the cross-language limitations of textual prompts. The preliminary mixed-language experiments in Chinese and English also align more closely with the applications of real-world speech recognition, offering a solution for complex multilingual mixed speech recognition.

## 4.5 Results on Flickr8K

In the context of the real-world speech dataset Flickr8K, the pre-trained Whisper model showcases a notable advantage in English speech recognition, surpassing the performance of most conventional models. These traditional models often conclude that incorporating visual cues fails to offer additional improvements. Ni et al. [31] suggests that this phenomenon occurs because speech serves as the primary input for ASR systems, with visual cues playing

**Table 3: Main WER(%) results on Flickr8K. The visual column indicates the use or non-use of visual cues.**

| Model | Visual | WER ↓ | CER ↓ |
|---|---|---|---|
| Sun et al. [44] | | 14.75 | / |
| Sun et al. [44] | ✓ | 13.81 | / |
| Srinivasan et al. [43] | | 13.60 | / |
| Srinivasan et al. [43] | ✓ | 14.10 | / |
| Oneață and Cucu [33] | | 3.80 | / |
| Oneață and Cucu [33] | ✓ | 4.30 | / |
| VILAS[31] | | 3.40 | / |
| VILAS[31] | ✓ | 3.40 | / |
| WHISPER | | 2.42 | 1.02 |
| CIEASR | ✓ | **2.05** | 0.89 |

merely an auxiliary role. When the speech input is sufficiently clear, the inclusion of other modalities might lead to interference, potentially impairing recognition capabilities.

As shown in **Table 3**, our model enhances the ASR capabilities built on the Whisper by integrating visual cues as soft prompts. Given that the visual component functions independently and does not contribute directly to the loss computation, it effectively serves as a soft prompt for speech decoding. This approach to cue integration minimizes the disruption caused by redundant multimodal information, emphasizing the supportive function of multimodal cues instead. Our findings offer fresh perspectives on cue integration techniques, promising valuable directions for future research in multimodal speech recognition.

## 4.6 Ablation Study on prefix tuning

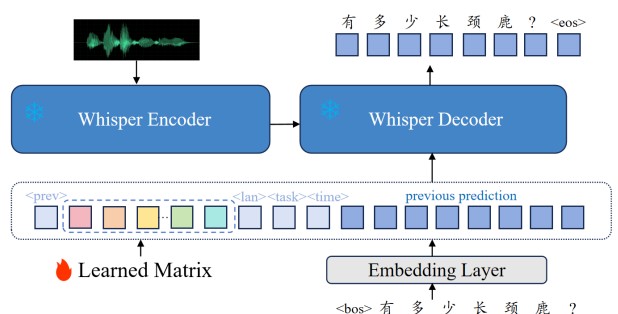

**Figure 3: Ablation study on learned prefix matrix: exploring the impact of trainable prefix on model performance.**

In our approach, we froze the Whisper model to ensure its original configuration remains unaltered. Recent studies [21, 24, 25] have indicated that adding a prefix to pre-trained models significantly enhances their performance, and our approach of incorporating an image soft prompt similarly introduces a prefix-like element.

To discern the unique contribution of contextual visual cues beyond the impact of prefix addition, we design an ablation study utilizing learnable parameters that mimic the form of the image

prompt but function as blank cues. This setup aims to neutralize the potential performance enhancements attributed solely to the prefix structure, allowing for a focused examination of the impact of visual cues. The rationale behind this method is to isolate and quantify the specific advantages that contextual images confer on speech recognition, thereby clearly distinguishing between the effects of prefix addition and the substantive enhancement provided by visual information. This approach enables a detailed analysis of the effectiveness of visual cues while keeping all other model hyperparameters and training conditions constant, as detailed in **Figure 3**.

Specifically, we define a learnable parameter matrix with the same shape as the Q-Former queries, which can be considered a blank image cue. This approach aims to isolate the effect of introducing this prefix matrix on the overall ASR performance of the Whisper model.

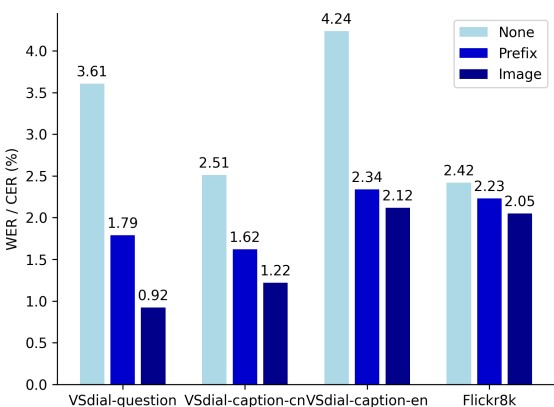

**Figure 4: Improvement of prefix and scene image cues. None denotes pure speech recognition, Prefix indicates the addition of a trainable prefix, and Image represents using scene image cues. The Chinese dataset (left half) uses CER(%), and the English dataset (right half) uses WER(%).**

Our experiments meticulously assess the impact of integrating contextual visual cues alongside the structural addition of a prefix to the Whisper model. The experimental results are presented in **Figure 4**.We can conclude that:

- The introduction of a prefix prompt has been observed to enhance the performance of speech recognition systems, particularly in synthetic speech datasets. Synthetic speech collections harbor more distinct speech domain features, which can be effectively captured and learned through the utilization of prompts. Similarly, although real-world speech datasets present a higher level of complexity, they also exhibit domain-specific characteristics, such as speakers' vocal habits and regional dialects, which still hold potential for further investigation in future research endeavors.
- Compared to prefix prompts, image prompts can further enhance performance, especially within Chinese speech datasets, as indicated by the first two columns in the table. This improvement can be attributed to the widespread presence of

homophones in Chinese. In **section 4.7**, we delve deeper into the corrective capabilities of image prompts concerning homophones.

Through the comparison with the blank image experiment, we observe its adaptability to the features of the speech dataset, which can enhance the ASR recognition accuracy to a certain extent.

## 4.7 Experimental Analyses on Noun and Pronoun Recognition

**Table 4: Total number of nouns contained and correctly identified in three test sets. Among them, VSDial includes 8k test samples, and Flickr8K includes 5k test samples.**

| Dataset | Cues | Correct | Total | Acc(%) ↑ |
|---|---|---|---|---|
| VSDial-question | none | 9743 | 10273 | 94.84 |
| VSDial-question | prefix | 10015 | 10273 | 97.48 |
| VSDial-question | image | 10119 | 10273 | **98.50** |
| VSDial-caption | none | 27457 | 28777 | 95.41 |
| VSDial-caption | prefix | 27682 | 28777 | 96.19 |
| VSDial-caption | image | 27959 | 28777 | **97.16** |
| Flickr8K | none | 15517 | 15958 | 97.24 |
| Flickr8K | prefix | 15512 | 15958 | 97.21 |
| Flickr8K | image | 15563 | 15958 | **97.52** |

**Table 5: Total number of pronouns contained and correctly identified in three test sets.**

| Dataset | Cues | Correct | Total | Acc(%) ↑ |
|---|---|---|---|---|
| VSDial-question | none | 1929 | 2252 | 85.73 |
| VSDial-question | prefix | 1980 | 2252 | 87.92 |
| VSDial-question | image | 2175 | 2252 | **96.58** |
| VSDial-caption | none | 380 | 435 | 87.36 |
| VSDial-caption | prefix | 385 | 435 | 88.51 |
| VSDial-caption | image | 392 | 435 | **90.11** |
| Flickr8K | none | 1006 | 1023 | 98.34 |
| Flickr8K | prefix | 1003 | 1023 | 98.04 |
| Flickr8K | image | 1000 | 1023 | 97.75 |

We investigate the contribution of contextual images to the recognition of specific types of vocabulary by analyzing the accuracy of noun recognition as well as pronoun recognition.

We use the Spacy model to identify nouns and pronouns in 8,000 samples each from the VSDial-question and VSDial-caption test sets and 5,000 samples from Flickr8K test sets. From this, we calculate the Noun Recognition Rate (NRR) and Pronoun Recognition Rate (PRR), with the results presented in **Table 4** and **Table 5**.

The VSDial-caption, being a comprehensive description of images, frequently utilizes numerals and thus contains more nouns

and fewer pronouns. Conversely, VSDial-question primarily poses questions about the most prominent entities in images, containing a higher frequency of personal pronouns and exhibiting stronger significance. In English, the phenomenon of homophonic nouns and pronouns is less common, hence the system's correction capability is not significant for the Flickr8K dataset.

The provision of contextual image cues in the VSDial-question dataset resulted in improvements of 3.66% for NRR and 10.85% for PRR; in the VSDial-caption dataset, these cues led to enhancements of 1.75% for NRR and 2.75% for PRR. Utilizing a prefix also significantly boosts performance, primarily in the case of abstract nouns, as well as demonstrative pronouns, relative pronouns, and indefinite pronouns, where images cannot provide auxiliary support.

To more vividly illustrate the impact of scene image cues, we delve into error examples from VSDial in the absence of such cues, highlighting the corrective potential of image cues. This analysis is clearly depicted in **Figure 5**.

The VSDial-question dataset exhibits a high correction rate for erroneous samples, achieving 75% for nouns and 82% for pronouns. Conversely, the VSDial-caption dataset shows correction rates of 50% for nouns and 36% for pronouns. This discrepancy arises because the question format directly inquires about the image, focusing more on the most prominent entities within it, and includes a greater number of personal pronouns that can be corrected. On the other hand, the caption format provides a comprehensive description of the image, prioritizing a thorough depiction of entities and tending to describe using numerals rather than pronouns.

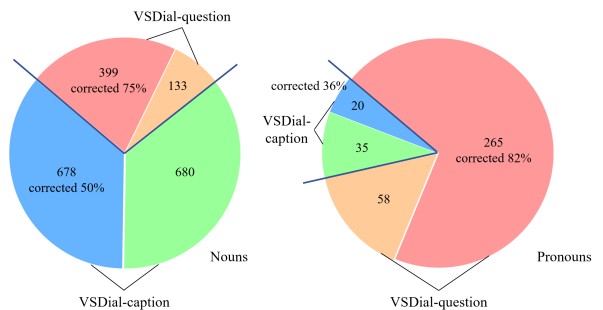

**Figure 5: The number of corrected nouns and pronouns in VSDial dataset. Warm colors represent error samples from VSDial-question, and red for corrections. Cool colors represent error samples from VSDial-caption, and blue for corrections.**

We can conclude that contextual images make a significant contribution to the identification and correction of entity information in homophone errors within the Chinese speech dataset. Our method offers a viable and rapid solution for the recognition of homophones, which can alleviate to some extent the high costs and inconvenience of artificial auxiliary cues in ASR systems.

## 4.8 Experimental Analyses with Activation Map

In order to further analyze our CIEASR model for enhanced transparency and interpretability, we have integrated the Grad-CAM[39]

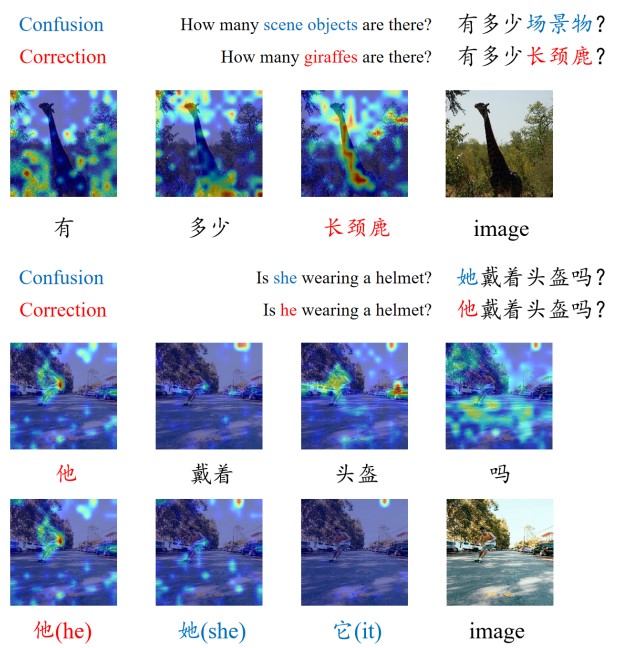

| Confusion | How many scene objects are there? | 有多少场景物? |
| Correction | How many giraffes are there? | 有多少长颈鹿? |

有   多少   长颈鹿   image

| Confusion | Is she wearing a helmet? | 她戴着头盔吗? |
| Correction | Is he wearing a helmet? | 他戴着头盔吗? |

他   戴着   头盔   吗

他(he)   她(she)   它(it)   image

**Figure 6: The visualization of activation maps between speech and the scene image.**

technique to examine the impact of image information on the probability of label predictions made by the whisper decoder. More precisely, we initially acquire the probability distribution $y^c$ for each word as predicted by the whisper decoder for the $c^{th}$ token, and compute the gradient of $y^c$ with respect to the final feature map $A$ extracted by the vision encoder, expressed as $\frac{\partial y^c}{\partial A}$. Subsequently, the gradients derived via backpropagation are averaged across each channel dimension for every pixel value—reminiscent of Global Average Pooling—to determine the significance weights for each channel, delineated by Equation 2:

$$\alpha_k^c = \frac{1}{Z} \sum_i \sum_j \frac{\partial y^c}{\partial A_{ij}^k} \quad (2)$$

Herein, $Z$ signifies the count of pixels within the feature map, and $A_{ij}$ symbolizes the pixel value at the $i, j$ position of the $k^{th}$ feature map.

Utilizing the importance weights ascertained heretofore, the channel features of the feature map are weighted accordingly, and through the application of a ReLU function, the activation map is procured, as demonstrated in Equation 3:

$$L_{\text{Grad-CAM}}^c = \text{ReLU}\left(\sum_k \alpha_k^c A^k\right) \quad (3)$$

The utilization of the ReLU function is intended to exclusively consider those pixels that positively influence the token $c$.

During the practical analysis, we eschewed token-level predictions in favor of utilizing the predicted probabilities of each individual word to construct the activation map. This methodology enabled us to scrutinize the impact of each pixel on the predictive

accuracy of every word. We curated a selection of images that exemplified the model's proficiency in correcting errors. By leveraging the Grad-CAM approach, we illustrated the critical image regions that were instrumental in the model's sequential word predictions.

As shown in Figure 6, the Chinese speech means "How many giraffes are there?" Without scene images, "giraffe" is confused with the phonetically similar "scene objects". When our scene images are presented, our model is adept at correcting errors to predict the correct word "giraffes". Furthermore, by analyzing the activation map, it becomes evident that the image region corresponding to the giraffes plays a pivotal role in the prediction of the word "giraffes", while this area does not attract attention during the prediction of other words. This indicates that the error correction capability of our model is attributed to the presence of giraffes in the image.

In a similar vein, in the second example, the completely identical pronunciation of third-person pronouns (he/she/it) in Chinese poses a challenge for pure speech recognition systems to determine the semantic gender based solely on the speech signal itself. However, the scene image provided offers visual cues about the person's gender, and it is observed that this specific region contributes significantly to the accurate prediction of the correct pronoun "he". In contrast, when predicting the incorrect pronouns "she" or "it", there is almost no attention paid to the image (in our system, the probabilities of "she" and "it" are extremely low).

The focus areas of the activation map provide excellent interpretability for our method of integrating scene images as soft prompts. The fusion of contextual visual information is profoundly effective, especially for homophone confusion errors involving entity nouns and pronouns.

## 5 CONCLUSION

In this study, we introduce CIEASR, a pioneering approach to ASR that leverages unsupervised pre-trained speech models and enhances them with visual contextual information, particularly scene images, to address the challenge of homophone discrimination—a task where traditional speech-only models falter. By integrating image semantic information as soft prompts, our model not only improves interpretability but also paves the way for innovative cue fusion methods in speech recognition. The efficacy of CIEASR is underscored by its achievement of state-of-the-art results on both VSDial and Flickr8K datasets. Our comprehensive analyses further validate that scene images significantly contribute to the precise recognition of entity nouns and Chinese personal pronouns with identical pronunciations. Moving forward, we will further investigate how to more effectively integrate an increased range of multimodal cues into speech recognition systems under the paradigms of pre-training and prompts, aiming to study the complementary mechanisms between multimodal information.

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
