# OpenReview forum: "CIEASR:Contextual Image-Enhanced Automatic Speech Recognition for Improved Homophone Discrimination"
_acmmm.org/ACMMM/2024/Conference — MM2024 Poster_

### Official Review · Reviewer_J7EA · 2024-05-20

**Rating:** 4
**Confidence:** 3

**Summary:**

This paper presents an advanced approach to improving speech recognition by integrating audio, visual, and semantic features to address homophone errors. The authors transform speech into a Mel spectrogram and extract acoustic features using the Whisper model's encoder. These features are enhanced through cross attention, blending auditory data with semantic information from scene images processed by the Q-Former model. The adapted Whisper decoder utilizes these image-derived soft prompts alongside text embeddings to generate accurate transcriptions. Evaluations on expanded VSDial and Flickr8K datasets show significant performance improvements.

**Strengths:**

1. **Clarity and Detail**:
   - Clear explanations of the methodology, including preprocessing steps, integration of multimodal data, and detailed experimental setups.

2. **Problem Statement and Motivation**:
   - Addresses the challenge of homophone errors in speech recognition, providing a strong motivation for integrating visual and semantic information.

3. **Innovative Contributions**:
   - Novel use of Q-Former for extracting semantic information from images and integrating it as soft prompts in the Whisper model.

4. **Evaluation**:
   - Comprehensive evaluation on both synthetic (VSDial) and real-world (Flickr8K) datasets, demonstrating significant improvements in CER and WER.

5. **Relevance**:
   - Practical implications for real-world speech recognition, particularly in multilingual and visually challenging environments.

**Limitations:**

1. **Computational Complexity**:
   - The integration and processing of multimodal data require significant computational resources, which may limit scalability and real-time application.

2. **Limited Generalization Discussion**:
   - The paper lacks a thorough discussion on how well the proposed method generalizes to diverse real-world conditions, such as various languages, accents, and spontaneous speech.

3. **Scope of Ablation Study**:
   - The ablation study focuses on visual prompts but does not explore the impact of different types of visual and semantic information, limiting the understanding of various influencing factors.

**Suitability:**

3

---

### Official Review · Reviewer_NB3V · 2024-05-23

**Rating:** 3
**Confidence:** 4

**Summary:**

The context images are applied to enhance the performance of the ASR task.

**Strengths:**

The scene image cues serve as soft prompts and integrates into pre-trained Whisper model, obtaining significant improvements in the recognition of entity nouns and pronouns.

**Limitations:**

In general, the proposed model architecture only integrates existing well-designed neural networks, please clarify the rationale of the model design related to the task in this work.
As the key in this work, the feature fusion (prompt process) across different modalities is required to be further clarified.
The comparative methods in this work are limited, please consider more recent works as baselines to validate the proposed approach.
The implementation details of the proposed approach, as well as the selective baselines, are also required to provide to improve the reproducibility.
In addition, more analysis is required to understand the work process of the proposed model.

**Suitability:**

2

---

### Official Review · Reviewer_r8mR · 2024-05-23

**Rating:** 3
**Confidence:** 3

**Summary:**

This paper introduces a novel multimodal speech recognition model CIEASR to correct homophone errors by incorporating scene images as soft prompts.

**Strengths:**

This paper is easily followed and provides a clear description of the methods.
A novel multimodal ASR model, called CIEASR, is introduced to solve the challenge of homophone discrimination in the ASR systems by incorporating contextual visual comprehension.
An expanded dataset VSDial-caption is conducted to support the experiments.

**Limitations:**

1. While the authors have introduced a new framework aimed at addressing the challenge of homophone recognition, it appears that all components of the framework are constructed based on existing technologies. There is a notable absence of technical innovation within the proposed system.
2. It is suggested to provide a detailed description of the fusion process between visual cues and speech signals. A comprehensive elucidation of this process is essential to enhance the perceived technical innovation of the work. Additionally, the term 'learned queries' mentioned in Figure 3 requires clarification.
3. Although the paper reports experiments conducted across three datasets, the comparative methods are limited. To strengthen the validity of the findings and to provide a more comprehensive evaluation of the proposed approach, it is recommended that the authors incorporate a broader range of comparative methods.
4. The manuscript currently lacks a dedicated section detailing the comparative methods, which is crucial for ensuring the fairness and transparency of the experimental evaluation. It is recommended that the authors include a section that thoroughly describes the comparative methods and the experimental setup.
5. Furthermore, the integration of textual information within the framework is not clearly explained. It would be beneficial for the authors to elucidate whether the incorporation of textual data is facilitated through a pre-trained model or another mechanism.
6. Table 1, according to the reviewers' understanding, the textual information is more direct than visual cues, implying that using text as a cue should also yield significant performance improvements. Furthermore, the combination of text and image encompasses richer information, therefore, it should obtain better results compared to the image cue. To ensure the validity of the experiments, it is imperative that the authors provide a detailed account of the fusion techniques and methodologies for the text and image+text cues.

**Suitability:**

3

---

### Official Review · Reviewer_qWny · 2024-05-24

**Rating:** 5
**Confidence:** 4

**Summary:**

The paper introduces CIEASR, a multimodal speech recognition model that utilizes scene images as soft prompts to correct homophone errors in Automatic Speech Recognition (ASR). It refines the VSDial dataset and demonstrates significant improvements in Character Error Rate (CER) on both VSDial and Flickr8K datasets.

**Strengths:**

1. The approach of using scene images as soft prompts is innovative and addresses the challenge of homophone discrimination effectively.
2. The paper is well-written, with clear explanations of the model architecture and the fusion strategy for integrating visual cues.

**Limitations:**

1. The paper could explore the model’s performance across a wider range of languages and dialects to ensure broader applicability.
2. While the paper claims state-of-the-art results, a more detailed comparison with existing methods, including quantitative metrics, would strengthen the evaluation.

**Suitability:**

3

---

### Meta-Review · Area_Chair_2CDK · 2024-07-03

**Recommendation:** Accept (Poster)
**Confidence:** 5

**Metareview:**

All reviewers agree that the proposed method is interesting and innovative. All reviewers comment that utilizing scene images as soft prompts to correct homophone errors in ASR is novel and contribute significantly to the increased performance of the presented approach. The reviewers commented that generalisation capability of the model should be further investigated (in terms of different languages, accents, etc.). The rebuttal addressed majority of these additionally raised questions. After reading the rebuttal, one of the reviewers upgraded the final score. One reviewer lowered their score but the additional concerns raised by that reviewer are not about any of the major quality points of the presented work but represent a further request for additional auxiliary explanations. The authors are therefore recommended to include a detailed explanation of computational complexity, diversity of the VSDial and Flickr8K datasets, and a discussion on how various multimodal inputs influence performance. Given a general appreciation of the work by the reviewers, I believe that the paper will be of interest to the audience attending ACM MM and would recommend a presentation of the work as a poster.